# Infection-Triggered Hyperinflammatory Syndromes in Children

**DOI:** 10.3390/children9040564

**Published:** 2022-04-15

**Authors:** Martina Rossano, Greta Rogani, Maria Maddalena D’Errico, Martina Cucchetti, Francesco Baldo, Sofia Torreggiani, Gisella Beretta, Stefano Lanni, Antonella Petaccia, Carlo Agostoni, Giovanni Filocamo, Francesca Minoia

**Affiliations:** 1Pediatric Rheumatology, Fondazione IRCCS Cà Granda Ospedale Maggiore Policlinico, Milano IT and University of Milan, 20122 Milan, Italy; martina.rossano@policlinico.mi.it (M.R.); greta.rogani@unimi.it (G.R.); martina.cucchetti@unimi.it (M.C.); francesco.baldo@unimi.it (F.B.); sofia.torreggiani@unimi.it (S.T.); gisellabeatriceberetta@gmail.com (G.B.); stefano.lanni@policlinico.mi.it (S.L.); antonella.petaccia@policlinico.mi.it (A.P.); carlo.agostoni@unimi.it (C.A.); francesca.minoia@policlinico.mi.it (F.M.); 2Department of Medical Sciences, IRCCS Casa Sollievo della Sofferenza, 71013 San Giovanni Rotondo, Italy; mm.derrico@gmail.com

**Keywords:** macrophage activation syndrome, infections, trigger, rheumatic diseases

## Abstract

An association between infectious diseases and macrophage activation syndrome (MAS) has been reported, yet the exact role of infection in MAS development is still unclear. Here, a retrospective analysis of the clinical records of patients with rheumatic diseases complicated with MAS who were treated in a pediatric tertiary care center between 2011 and 2020 was performed. Any infection documented within the 30 days preceding the onset of MAS was reported. Out of 125 children in follow-up for systemic rheumatic diseases, 12 developed MAS, with a total of 14 episodes. One patient experienced three episodes of MAS. Clinical and/or laboratory evidence of infection preceded the onset of MAS in 12 events. Clinical features, therapeutic strategies, and patient outcomes were described. The aim of this study was to evaluate the possible role of infection as a relevant trigger for MAS development in children with rheumatic conditions. The pathogenetic pathways involved in the cross-talk between uncontrolled inflammatory activity and the immune response to infection deserve further investigation.

## 1. Introduction

Macrophage activation syndrome (MAS) is a rare and severe complication of rheumatic diseases that occurs due to the activation of the monocyte-macrophage system and leads to various degrees of coagulopathy, pancytopenia, unremitting fever, lymphadenopathy, and liver and central nervous system dysfunction. This explains why MAS may become a potentially life-threatening complication. First described in children with systemic juvenile idiopathic arthritis (sJIA), MAS has been subsequently reported in other rheumatic conditions in children, such as juvenile dermatomyositis (JDM), Kawasaki disease (KD), and childhood systemic lupus erythematosus (cSLE) [1,2,3]. In 2005, Ravelli et al. presented the first set of preliminary diagnostic criteria for MAS in patients with sJIA [4]. In addition, in 2016, EULAR/ACR/PRINTO proposed an update of the aforementioned criteria for MAS in patients with sJIA [3].

The most common biochemical features of MAS are increased levels of ferritin, triglycerides, liver enzymes, lactate dehydrogenase, and C-reactive protein. Other laboratory findings include low levels of fibrinogen, cytopenia, and a reduction in erythrocyte sedimentation rate [5,6]. Although hemophagocytosis can be observed in about 60% of bone marrow examinations performed during MAS, this phenomenon is not mandatory for diagnosis since it can be absent, especially in the initial stages of MAS [7]. It was reported in the largest published cohort of patients with sJIA that the main causative factor for developing MAS is the active underlying disease. It is of interest to note that in this study, 34.1% of patients experienced an infection at the onset of MAS. This suggests that infections may play a pivotal role in inducing MAS. Different microorganisms have been reported as potential triggers for MAS, such as the Epstein–Barr virus, the cytomegalovirus (CMV), other herpes viruses [8,9], and the influenza virus [10], in addition to other bacterial, parasitic, and fungal infections. More recently, SARS-CoV-2 has also been associated with MAS development [11]. Although drug-induced MAS has also been postulated, the causal relation of this phenomenon still needs to be explored with great caution, since drug changes occur routinely in the setting of patients with a highly active underlying disease [12,13,14,15].

The aim of this study was therefore to evaluate the prevalence of MAS induced by infectious events in a 10-year retrospective case series of a tertiary pediatric care center.

## 2. Materials and Methods

The clinical data from the 30 days preceding the onset of MAS in a cohort of patients with rheumatic diseases followed in our center between 2011 and 2020 were reviewed retrospectively. Data collection included clinical and laboratory findings, treatment decision making, and outcomes. All patients with sJIA fulfilled the 2016 classification criteria for MAS in sJIA patients [4]. Patients with sJIA who developed MAS before 2016 were diagnosed according to the 2005 preliminary diagnostic guidelines for the diagnosis of MAS in sJIA patients [3]. In patients with cSLE, the diagnosis of MAS was performed using the preliminary diagnostic guidelines for MAS developed for patients with cSLE [14]. 

In patients with JDM, the diagnosis of MAS was carried out by taking into account the aforementioned laboratory and clinical findings typical of patients with MAS, due to the current lack of a validated set of classification criteria for MAS in JDM patients. Interestingly, these patients fulfilled the 2016 classification criteria for MAS in sJIA patients [4,15].

Although multisystem inflammatory syndrome in children (MIS-C) secondary to SARS-CoV-2 infection is considered as part of the wide spectrum of cytokine storms, its pathogenesis and especially its relation to hemophagocytic syndromes are still not completely understood. Thus, in order to reduce the heterogeneity of our cohort, children with MIS-C were excluded from enrollment. For the purpose of our study, we decided to recruit patients with clinically evident infections irrespective of the detection of the causative agent in laboratory findings. Descriptive statistics were reported in terms of medians and ranges for continuous variables and in terms of absolute frequencies and percentages for categorical variables.

## 3. Results

Between January 2011 and December 2020, we followed 125 children with systemic rheumatic diseases: 27, 59, and 39 patients had JDM, jSLE, and sJIA, respectively. In this period, we observed 14 episodes of MAS in 12 patients (Table 1). 

Overall, eight patients (66.7%) were female. The median age at MAS onset was 11.75 years (IQR 3.97–12.48).

The median duration of the rheumatic underlying disease prior to MAS was 2.15 years (IQR 0.06–2.75). 

Six patients who developed MAS had sJIA, two had JDM, three had cSLE, and one had undifferentiated arthritis (UA).

The patient with UA was a female (case 1) who developed three episodes of MAS throughout the disease course. At disease onset, she had clinical features resembling sJIA, such as a high spiking fever, erythematous rash, serositis, and arthritis. Her first episode of MAS was well controlled with a high dose of steroids, cyclosporine, and intravenous immunoglobulins. The subsequent episode of MAS occurred two and a half years later, when the patient discontinued cyclosporine. After 6 years, she experienced a third episode of MAS due to a urinary tract infection, which was coupled with a disease flare characterized by fever, acute anterior uveitis, and sacroileitis on an MRI. A genetic analysis performed after the last episode of MAS showed a mutation in the PRF1 gene (c.1262T > G, p.F421C) [16]. 

Of the remaining patients who developed MAS, in five cases MAS occurred within the first month from the onset of the underlying rheumatic disease.

In 10 cases, causative microorganisms were identified using the laboratory work-up carried out during hospitalization. Conversely, in two cases, the infection was diagnosed on the bases of clinical findings alone, which can be described as follows: 

One patient (case 7) had vomiting and diarrhea, which suggested the presence of acute gastroenteritis; one patient (case 1c) exhibited strangury along with hematuria, proteinuria, and leukocyturia in a urine analysis, which were consistent with a urinary tract infection. 

Two patients presented with full-blown MAS after the introduction of a new drug (sulphasalazine in case 2 and mycophenolate mofetil in case 12). No trigger was identified in one patient (case 6).

Bone marrow aspiration was performed in nine cases and showed evidence of bone marrow hemophagocytosis in five of them (55.6%). 

The most common clinical feature was unremitting fever >38 °C (13 cases); eight subjects showed serositis (pericardial effusion, pleural effusion, or both). 

All patients were hospitalized and were treated with intravenous pulses of methylprednisolone for 3–5 days (10–30 mg/kg/day; max 1000 mg/day) and cyclosporin A; intravenous immunoglobulins were added in four cases. An anti-IL-1 drug (anakinra) was used in three cases at a starting dose of 5 mg/kg/days and then tapered according to the clinical and laboratory response. 

A transfusion of frozen plasma was added in seven cases (1a,1c, 2, 3, 5, 8, and 9) and blood transfusions were given in five episodes (1c, 2, 3, 5, and 9). 

Overall, 11 patients were successfully treated for MAS and did not develop any clinical sequela. Only one patient (case 3) developed spastic tetraplegia. This patient was admitted after 5 days of unremitting fever for sepsis from *Staphylococcus hominis*. A brain hemorrhage occurred before the patient was diagnosed with MAS. 

## 4. Discussion

The current study aimed to evaluate the prevalence of MAS induced by infectious events in a cohort of patients followed in a single pediatric tertiary care center for rheumatic diseases. In our sample, around 11% of rheumatic patients developed MAS, with a total of 14 events. It is of interest to note that in the majority of these cases, MAS was preceded by an infection. 

MAS often complicates rheumatic diseases at the onset or during the exacerbation of the underlying disorder [17], and the plausible role of infections as a co-trigger has been previously reported in literature. The first description of MAS was reported in seven subjects affected by sJIA in 1985; in this sample, the possible causative association with infections [18] was postulated. Furthermore, in 2001, Shawney et al. presented a retrospective review of nine cases of MAS in children with underlying rheumatic conditions; an infectious trigger was suggested in eight cases a few weeks before developing MAS [19], with the definite identification of a specific infectious agent in four cases. 

The clinical presentation of MAS is common to other cytokine storm syndromes in the spectrum of hemophagocytic lymphohistiocytosis (HLH) [20] and is believed to represent the same end-stage pathophysiological state reached by different initiating triggers of uncontrolled inflammation [21]. Hemophagocytic syndromes are classified as primary forms, including genetic defects in cytotoxicity leading to familial hemophagocytic lymphohistiocytosis (FHL) and immunodeficiency-associated hyperinflammatory syndromes, and secondary forms, which comprise MAS, infection-related HLH, and malignancy-related HLH. Although the pathogenetic mechanisms underlying the secondary forms of HLH are not yet fully understood, a growing body of evidence suggests that MAS development is driven by a convergence of impaired cytotoxic function and cytokine overproduction [22,23,24]. 

A mainstay of the body’s defense against viral infections and tumors is represented by natural killer (NK) cells [25]. The temporary impairment of perforin expression and cytotoxic function has already been reported in patients with sJIA, especially during MAS [26,27,28,29]. Furthermore, in sJIA-associated MAS, heterozygous mutations in genes affecting cytotoxic function have been detected [21]. In accordance with this evidence, the threshold or multilayer model for MAS pathogenesis was proposed, in which genetic predisposition, disease activity, and other triggers, such as infections, contribute in different proportion to a massive cytokine release, leading to the common final end-stage of full-blown MAS [21,22,23,24,25,26,27,28,29,30,31]. Interestingly, recent studies have supported the idea of an intriguing cross-talk between Toll-like receptors and type 1 interferon responses, typically involved in the response to infections, and the overproduction of IL-18, a pivotal cytokine in MAS pathogenesis [32]. A comparison between MAS and MIS-C is far beyond the scope of this paper; however, future studies on the pathogenetic model of SARS-CoV-2 related cytokine storms will likely lead to important insights in the understanding of the interplay between infection and the development of MAS [30]. 

The retrospective nature of this case series is one of its main limitations. Another caveat of our study is the fact that the causative microbiological agent was not found in all cases of MAS thought to be induced by infectious events. Furthermore, except for in patients with cSLE, validated criteria to classify MAS in other rheumatic conditions, such as JDM and non-sJIA, are not available so far. Due to the current lack of diagnostic criteria for MAS, physician’s integration between clinical and laboratory findings is still considered as key for the diagnosis of this complication.

## 5. Conclusions

The current study suggests that the frequency of MAS-preceding infectious events may be underestimated and supports the role of infection as a co-trigger for MAS development. Whether the early treatment of infections improves the outcomes in these patients is yet to be clarified. The limited size and high heterogeneity of our cohort does not allow us to draw conclusions about this issue. However, early targeted antimicrobial therapy should be initiated whenever possible. As a matter of fact, the MAS pathophysiology itself suggests that early pathogen clearance can reduce the inflammatory burden and the release of cytokines [33]. The pathogenetic pathways involved in the cross-talk between uncontrolled inflammation and the host’s immune response to infection deserve further investigation. 

## Figures and Tables

**Table 1 children-09-00564-t001:** Demographic characteristics, clinical features, and treatment of the 12 patients and pathogens identified.

Patient	Gender	Rheumatic Disease	Median Age at the Onset of Rheumatic Disease (Years)	Disease Duration Pre-MAS	Suspected Infectious Triggers	Clinical Features	Treatment
1	F	UA	10.75	0.05	(1a) *Clostridium difficile* (fecal test)	Persistent fever, hepatomegaly, motor polyneuropathy, sierositis, acute renal failure	High doses of methylprednisolone, ciclosporin A, intravenous immunoglobulins, metronidazole (IV)
				2.75	(1b) Coxsackie virus * (positive IgM)	Persistent fever, splenomegaly, pericardial effusion	High doses of methylprednisolone, ciclosporin A
				6.25	(1c) UTI	Persistent fever, serositis, acute renal failure	High doses of methylprednisolone, ciclosporin A, ampicillin-sulbactam IV
2	F	sJIA	3	11.5	No infectious trigger identified	Persistent fever, splenomegaly, pericardial effusion, petechiae, acute renal failure	High doses of methylprednisolone, ciclosporin A, anakinra
3	M	sJIA	7.3	2.9	*Staphylococcus hominis* (blood cultures)	Persistent fever, lymphadenopathy, brain hemorrhage, interstitial lung disease	High doses of methylprednisolone, cyclosporin A, intravenous immunoglobulins.
4	F	sJIA	14.25	2.75	*Entamoeba dispar/histolytica and**Endolimax nana* (parasitological search of feces positive)	Persistent fever, hepatomegaly, lymphadenopathy, pericardial effusion	High doses of methylprednisolone, cyclosporin A
5	M	sJIA	3	0.5	VZV (specific IgM)	Persistent fever, hepatosplenomegaly, interstitial lung disease	High doses of methylprednisolone, cyclosporin A, intravenous immunoglobulins, acyclovir IV
6	F	sJIA	10	0	No infectious trigger identified	Persistent fever, hepatomegaly, lymphadenopathy, pericardial effusion	High doses of methylprednisolone, cyclosporin A, anakinra
7	M	sJIA	2.5	0.25	Gastroenteritis	Persistent fever, hepatomegaly, lymphadenopathy,	High doses of methylprednisolone, cyclosporin A, anakinra
8	F	JDM	13.2	0.8	EBV (specific IgM)	Persistent fever, hepatosplenomegaly, serositis	High doses of methylprednisolone, cyclosporin A, intravenous immunoglobulins
9	F	JDM	4.3	0.1	Group A Streptococcus (throat swab)	Persistent fever, hepatosplenomegaly, interstitial lung disease, acute renal failure	High doses of methylprednisolone, cyclosporin A, intravenous immunoglobulins
10	F	SLE	12.25	0.05	Adenovirus * (specific IgM)	Persistent fever, hepatosplenomegaly	High doses of methylprednisolone, cyclosporin A
11	F	SLE	16.6	2.2	*Pseudomonas aeruginosa* (positive cultures from a cutaneous swab)	Persistent fever, hepatosplenomegaly	High doses of methylprednisolone, cyclosporin A, intravenous immunoglobulins,
12	M	SLE	10.2	0.05	Adenovirus (positive DNA)	Splenomegaly, serositis	High doses of methylprednisolone, cyclosporin A

UA—undifferentiated arthritis; sJIA—systemic juvenile idiopathic arthritis; JDM—juvenile dermatomyositis; SLE—systemic lupus erythematosus; UTI—urinary tract infection; VZV—varicella zoster virus; EBV—Epstein–Barr virus. * Presence of infection demonstrated by specific IgM.

## Data Availability

The data presented in this study are available on request from the corresponding author.

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
