# Peer review of "Infection-Triggered Hyperinflammatory Syndromes in Children"

_children, 2022, doi:10.3390/children9040564_

Round 1

Reviewer 1 Report

This is an interesting retrospective case series of consecutive patients with rheumatic diseases presenting to a rheumatology center with MAS looking at potential infectious triggers to the MAS episodes. 

Methods:

The authors state:

Systemic JIA associated-MAS occurred before 2016 were diagnosed according to the 2005 preliminary diagnostic guidelines for the diagnosis of MAS in JIA[2]; all patients with MAS in sJIA satisfied the 2016 classification criteria [3].

  • So all patients with sJIA met the 2016 criteria as well as the 2005 guidelines? If so, the wording needs to be clearer.
  • Also, although guidelines to diagnose SLE patients with MAS is defined, there are patients with JDM. How were they diagnosed?

Results:

  • Ref 17 is cited in a sentence about the undifferentiated JIA patient with recurrent MAS and it doesn’t seem to fit.
  • Also, what are the details of the patient with undifferentiated JIA? Does this patient have sJIA with other features or missing features that make diagnosis difficult according to ILAR? Most patients with recurrent MAS are severe refractory sJIA cases. It would be of interest to note what the patient’s clinical features were.
  • Table 1:
    • It would be helpful to have the disease duration at time of MAS in a separate column (age at MAS seems less important than this)
    • Also, please add the clinical manifestations of MAS including laboratory parameters
    • Is there a reason why the order of the patients is in this format? I would be helpful to have the diseases grouped together.
    • Although there is mention of the details of the infections in the text, I think readers would appreciate seeing this information in the table. For example: C. Difficile (I assume in the stool), viruses (IgM or viral culture?), Gp A Strep (pharyngitis, bacteremia, culture without symptoms?), etc. This would obviate the need for the paragraph describing the infections but would keep the paragraph with the unusual circumstances
    • Could vomiting and diarrhea could be due to MAS rather than gastroenteritis?
    • Although it is stated that no patient had COVID infection documented, were there cases of MAS following COVID? If not, one could just state that there were no cases of MAS following SARS CoV2 infection, which would be more straightforward.
  • The total N of each of the diseases seen in the center would be helpful to get a sense of the incidence of MAS
  • Table 2:
    • Not sure this table is needed, if you are listing clinical manifestations of MAS in table 1
    • If keeping, listing diseases in the last column (not just patient number) would be helpful to see whether there are differences in presentation
    • Is there a reason why the treatment and outcomes of the patients were not discussed? I think the readers would be interested. Did they all recover? Was there organ damage? It is a small cohort, but was there a difference between those that had an infectious trigger and those that did not?

Discussion

  • In an attempt to compensate for this, we tried to distinguish infections supported by positive cultural examinations or clear clinical signs from those suspected on the basis of serological tests
    • Should be “supported by positive microbiological cultures or clear clinical…”
  • criteria used to diagnose MAS have been validated only in patients with sJIA; however, so far physician’s clinical diagnosis is still considered the gold standard for the diagnosis of MAS in rheumatologic conditions other than sJIA
    • Please elaborate and explain why this is the case (I would argue that this is the case in sJIA also, compared to HLH). This is important to be discussed in a general pediatric journal

Conclusions

  • Is there a statement that could be made whether this relationship between infection and MAS means something clinically? For example, is there any evidence that early detection of infection and treatment decrease the chance of MAS? If not, it should be mentioned that this is the case.

Reviewer 2 Report

General comments:

The authors reported the role of infection as a co-trigger for MAS development. The mechanisms are still unknown adequately, however, they documented  their retrospective analysis on their experiences. The report is interesting with some further improvements.

Specific comments:

  1. Why does MAS occur frequently in rheumatic diseases? In adult onset Still's disease, it is observed the same situation.
  2. Are there any correlations between pathogens and clinical symptoms?
  3. In Case 1, MAS arose three times. Does the patient have any characteristics for easy susceptibility?
  4. What is the true trigger in MAS regardless of causative agents, that is, infection, underlying disease activity, a new drug, and so on?
  5. I am very interested in the therapeutic options in MAS. Could you please tell about that? How about IL-6 inhibitors?
  6. In your article, which drug did induce MAS?

Round 2

Reviewer 1 Report

I think the manuscript has been improved substantially, but there are a few comments:

  1. In the abstract, the following statement should be clarified:

    Between 1st January 2011 and 31st December 2020, on 78 children with new onset of systemic

    rheumatic diseases, tTwelve children (eight females and four males) developed MAS (one of them

    experienced 3 episodes.

    The total number of patients followed with rheumatic diseases (not just new onset) should be the denominator, not just new diagnoses, as MAS can occur at any time during the disease course. Also, it would be clearer to state (as you do later in the manuscript) that 14 episodes of MAS occurred in 12 patients.
  2. In materials and methods, this sentence is unclear:

    theIn patients with JDM the diagnosis of MAS was performed according

    to the upper-mentioned laboratory and clinical characteristics of the patients, since a

    validated classification criteria for MAS in JDM is available so far. Interestingly, those

    patients fulfilled the 2016 classification criteria for MAS in sJIA,

    What is the classification referred to regarding MAS in JDM? It is not stated anywhere. Secondly, were these actually validated criteria, or just proposed? I can only find reference to a series of JDM patients with possible MAS.
  3. In results, as stated above, it would be more useful to know the total number of patients seen at this center with rheumatic diseases during this time period rather than just new onset.
  4. In results, although the MAS treatments are described, antimicrobial treatments are not discussed. It would be useful to add to the table.
  5. Also, a patient is said to have a UTI, but how was that diagnosis made?

Author Response

Milan, April 8th

Ms. Francesca Tian

Assistant Editor, Children

Ref: children-1643668– Rossano M et al. Infections and macrophage activation syndrome in rheumatic diseases. A retrospective analysis in a tertiary pediatric care center and review of the

literature.

Dear Ms Tian,

Thank you for your kind support during this review process. We are pleased that the , after the first round of reviews, was considered potentially interested for Editors and reviewers.

We have upload to the websites a second version of the paper modified accordingly to the suggestions of the reviewer, we have accepted the previous changes and underlined the new one with track changes.

Here below we included our point-by-point responses to the matters and the adjustments that we have made in the manuscript consequently.

REVIEWER 1:

I think the manuscript has been improved substantially, but there are a few comments:

  1. In the abstract, the following statement should be clarified:

Between 1st January 2011 and 31st December 2020, on 78 children with new onset of systemic rheumatic diseases, tTwelve children (eight females and four males) developed MAS (one of them experienced 3 episodes.

The total number of patients followed with rheumatic diseases (not just new onset) should be the denominator, not just new diagnoses, as MAS can occur at any time during the disease course. Also, it would be clearer to state (as you do later in the manuscript) that 14 episodes of MAS occurred in 12 patients.

R: Thank to the reviewer for this suggestion. We modified the abstract accordingly.

  1. In materials and methods, this sentence is unclear: the In patients with JDM the diagnosis of MAS was performed according to the upper-mentioned laboratory and clinical characteristics of the patients, since a validated classification criteria for MAS in JDM is available so far. Interestingly, those patients fulfilled the 2016 classification criteria for MAS in sJIA,

What is the classification referred to regarding MAS in JDM? It is not stated anywhere. Secondly, were these actually validated criteria, or just proposed? I can only find reference to a series of JDM patients with possible MAS.

R: we apologize for the typing mistake, the word “not” was missing. The correct and modiefied sentence is: “In patients with JDM the diagnosis of MAS was performed carried out according to the upper-mentionedtaking into account the aforementioned laboratory and clinical characteristics typical findings of the patients with MAS, due to the current lack of since a validated set of classification criteria for MAS in JDM. Interestingly, those patients fulfilled the 2016 classification criteria for MAS in sJIA”

The text was corrected.

  1. In results, as stated above, it would be more useful to know the total number of patients seen at this center with rheumatic diseases during this time period rather than just new onset.

R: Thank you for the suggestion, the total number of patients followed between 2011 and 2020 were reported.

  1. In results, although the MAS treatments are described, antimicrobial treatments are not discussed. It would be useful to add to the table.

R: Table was modified accordingly

  1. Also, a patient is said to have a UTI, but how was that diagnosis made?

R: The diagnosis of UTI was suspected due to evidence of fever, strangury, presence of neutrophilic leucocytosis, elevated acute fase reactants, with a urine test positive for hematuria, proteinuria and leukocyturia; a urine culture was not reliable because the patient was already on antibiotic therapy at urinalysis. Blood and urine culture turned out negative. The response to antibiotics (ampicillin-sulbactam) was good, fever disappeared and lab examination progressively normalized. The patient was discharged after 15 days with oral antibiotics.  Three days later she was admitted again because of reoccurrence of fever, arthralgia and increased acute phase reactants. In the next few days, the patient developed a full-blown picture of MAS with hyperferritemia (up to 19.322 ng/ml), hypertransaminasemia, hypertriglyceridemia, hyponatremia, reduction in platelet counts and hypofibrinogenemia.

We detailed the in the text more concisely.

Concerning the extensive English revisions we addressed this issue having the manuscript checked by an English-speaking colleague.

We want to thank the reviewer for these valuable suggestions, this revisions has certainly improve our work, and wish that after this full revision the manuscript is suitable for publication in Children.

Kind regards

Yours sincerely,

Giovanni Filocamo, MD

(on behalf of the authors)

This manuscript is a resubmission of an earlier submission. The following is a list of the peer review reports and author responses from that submission.

Round 1

Reviewer 1 Report

The study is interesting. It's clear and well-written.  There are some controversy about the triggers in MAS patients and this paper contributes to underline the role of infections in rheumatological patients.

I suggest to change the conclusion of this study because it's  only a hypothesis and NK dysfunction in MAS patient is not analyzed in the study. It could be included in the discussion and a new conclusion based on the results should be included.

All references are prior to 2017. Some publications of the last years should be included in the bibliography.

In the table of triggers, the microbiological results and a drug (sulphasalazine) are included. Drugs as triggers are not well detailed in the study and probably other drugs were used in this group of MAS patients. I propose to omit this drug in the table and/or include a paragraph with the impact of drugs as MAS triggers in rheumatological patients.  

Author Response

Milan, 02 February 2022

Ms. Francesca Tian

Assistant Editor, Children

Ref: Children-1557593– Rossano M et al. Infections and macrophage activation syndrome in rheumatic diseases. A retrospective analysis in a tertiary pediatric care centre and review of the

literature.

Dear Ms Tian,

Thank you for your kind letter, regarding the editorial decision about our manuscript. We are delighted that the manuscript was considered of potential interest by Editors and reviewers. We have submitted in the journal website a revision of the manuscript that addresses all issues raised by reviewers. We list below are our point-by-point responses to the issues and the modifications that we have made in the manuscript accordingly.

Reviewers' comments:

Reviewer #1: The study is interesting. It's clear and well-written.  There are some controversy about the triggers in MAS patients and this paper contributes to underline the role of infections in rheumatological patients.

I suggest to change the conclusion of this study because it's only a hypothesis and NK dysfunction in MAS patient is not analyzed in the study. It could be included in the discussion and a new conclusion based on the results should be included.

R: We thank the reviewer for the suggestion. The paragraph on NK dysfunction has been included in the discussion and conclusions have been modified in the text accordingly.

All references are prior to 2017. Some publications of the last years should be included in the bibliography.

R: We have updated references including recent relevant papers published in the last years (see Introduction, 3rd paragraph and Discussion, 3rd and 4th paragraphs). 

In the table of triggers, the microbiological results and a drug (sulphasalazine) are included. Drugs as triggers are not well detailed in the study and probably other drugs were used in this group of MAS patients. I propose to omit this drug in the table and/or include a paragraph with the impact of drugs as MAS triggers in rheumatological patients. 

R: We thank the reviewer; the table has been modified accordingly to the suggestion

We wish to thank the reviewers for all their helpful comments, which have led us to improve our manuscript, and hope that in its revised version the manuscript is suitable for publication in Children.

Yours sincerely,

Giovanni Filocamo, MD

(on behalf of the authors)

Reviewer 2 Report

Including a larger number of children with the same disease may aid in extrapolating more factors that affect the occurrence of MAS.

There is no need to present tables 1 and 2 for the MAS criteria. The corresponding references are enough.

Consider round brackets for the following: cytomegalovirus [CMV]

Again it is emphasized that there is a potential role of disease activity in the triggering of MAS. Recruiting a large number of cases with the same disease would make it easier to assess and analyze the contribution of the activity.

In order to evaluate the prevalence of infections preceding the onset of MAS and to evaluate the causative agents in our population, we retrospectively…

COMMENT: It is not appealing to speak of prevalence with 12 children. The total number of files was not mentioned. The work 'frequency' would be more appealing.

COMMENT: It is not appealing to over use the words 'we' and 'our'. Consider indirect speech e.g. the present work, the current cases….etc

As the study extended to 2020, it would be interesting to mention the COVID-19 status.

It is not familiar to cite references in the results section !

More detailed and indepth statistical analysis tests should be run to better interpret the results.

Author Response

Milan, 02 February 2022

Ms. Francesca Tian

Assistant Editor, Children

Ref: Children-1557593– Rossano M et al. Infections and macrophage activation syndrome in rheumatic diseases. A retrospective analysis in a tertiary pediatric care centre and review of the

literature.

Dear Ms Tian,

Thank you for your kind letter, regarding the editorial decision about our manuscript. We are delighted that the manuscript was considered of potential interest by Editors and reviewers. We have submitted in the journal website a revision of the manuscript that addresses all issues raised by reviewers. We list below are our point-by-point responses to the issues and the modifications that we have made in the manuscript accordingly.

Reviewer #2:

Including a larger number of children with the same disease may aid in extrapolating more factors that affect the occurrence of MAS.

R: We agree with the reviewer, but due to the rarity of MAS our cohort reflects the real-life clinical practice of a single centre, and its limited size does not allow any further speculation on other factors involved in MAS occurrence. This point has been specified as a limit of the study in the dedicated paragraph of the Discussion (see Discussion, last paragraph).

There is no need to present tables 1 and 2 for the MAS criteria. The corresponding references are enough.

R: We agree with the reviewer, tables 1 and 2 have been removed accordingly to the suggestion

Consider round brackets for the following: cytomegalovirus [CMV]

R: Done.

Again, it is emphasized that there is a potential role of disease activity in the triggering of MAS. Recruiting a large number of cases with the same disease would make it easier to assess and analyze the contribution of the activity.

R: please refer to the first point for the answer. Unfortunately, the small size and the heterogeneity of our monocentric cohort limited further speculations, as the evaluation of the role of disease activity which is beyond the scope of the paper. This point has been specified as a limit of the study in the dedicated paragraph of the Discussion (see Discussion, last paragraph).

“In order to evaluate the prevalence of infections preceding the onset of MAS and to evaluate the causative agents in our population, we retrospectively…”

COMMENT: It is not appealing to speak of prevalence with 12 children. The total number of files was not mentioned. The work 'frequency' would be more appealing.

COMMENT: It is not appealing to over use the words 'we' and 'our'. Consider indirect speech e.g. the present work, the current cases….etc

R: we modified the text accordingly to the suggestion.

As the study extended to 2020, it would be interesting to mention the COVID-19 status.

R: we thank the reviewer for the suggestion. In none of our patients SARS-CoV-2 has been documented at MAS onset. We included this data in the Results and commented on COVID19 both in Introduction and Discussion (see text modified accordingly).

It is not familiar to cite references in the results section!

R: we removed references from the Results section

More detailed and indepth statistical analysis tests should be run to better interpret the results.

R: We agree with the reviewer, unfortunately as abovementioned the small size and the heterogeneity of our monocentric cohort limited in-depth statistical analysis and further speculation between MAS subcategories. This point has been specified as a limit of the study in the dedicated paragraph of the Discussion (see Discussion, last paragraph).

We wish to thank the reviewers for all their helpful comments, which have led us to improve our manuscript, and hope that in its revised version the manuscript is suitable for publication in Children.

Yours sincerely,

Giovanni Filocamo, MD

(on behalf of the authors)

Round 2

Reviewer 2 Report

Thank you for making the required changes. However, the different diseases and low number is a major issue.